# Genome wide DNA methylation profiling identifies specific epigenetic features in high-risk cutaneous squamous cell carcinoma

**David Hervás-Marín**[1]☯, **Faatiemah Higgins**[2,3]☯, **Onofre Sanmartín**[4,5], **Jose Antonio López-Guerrero**[6], **M. Carmen Bañó**[2,3], **J. Carlos Igual**[2,3]*, **Inma Quilis**[2,3]*, **Juan Sandoval**[7]*

**1** Department of Biostatistics, Instituto de Investigación Sanitaria La Fe, Valencia, Spain, **2** Estructura de Recerca Interdisciplinar en Biotecnologia i Biomedicina (ERI BIOTECMED) Universitat de València, Burjassot, Valencia, Spain, **3** Departament de Bioquímica i Biologia Molecular, Universitat de València, Burjassot, Valencia, Spain, **4** Dermatology Department, Instituto Valenciano de Oncología, Valencia, Spain, **5** Facultad de Medicina, Universidad Católica de Valencia, Valencia, Spain, **6** Laboratory of Molecular Biology, Instituto Valenciano de Oncología, Valencia, Spain, **7** Biomarkers and Precision Medicine Unit (UByMP), Instituto de Investigación Sanitaria La Fe, Valencia, Spain

☯ These authors contributed equally to this work.
* jcigual@uv.es (JCI); inmaculada.quilis@uv.es (IQ); epigenomica@iislafe.es (JS)

**Data Availability Statement:** The data are deposited in the Array Express (EMBL-EBI) repository. Accession number: E-MTAB-8542.

## Abstract

Cutaneous squamous cell carcinoma (cSCC) is the second most common skin cancer. Although most cSCCs have good prognosis, a subgroup of high-risk cSCC has a higher frequency of recurrence and mortality. Therefore, the identification of molecular risk factors associated with this aggressive subtype is of major interest. In this work we carried out a global-scale approach to investigate the DNA-methylation profile in patients at different stages, from premalignant actinic keratosis to low-risk invasive and high-risk non-metastatic and metastatic cSCC. The results showed massive non-sequential changes in DNA-methylome and identified a minimal methylation signature that discriminates between stages. Importantly, a direct comparison of low-risk and high-risk stages revealed epigenetic traits characteristic of high-risk tumours. Finally, a prognostic prediction model in cSCC patients identified a methylation signature able to predict the overall survival of patients. Thus, the analysis of DNA-methylation in cSCC revealed changes during the evolution of the disease through the different stages that can be of great value not only in the diagnosis but also in the prognosis of the disease.

## Introduction

Nonmelanoma skin cancer (NMSC) is the most common cancer in humans and its incidence has risen dramatically [1]. Approximately 20% of nonmelanoma skin cancers are cutaneous squamous cell carcinomas (cSCC). cSCC is by far the most common cancer with metastatic potential in white populations and there is a progressive increase of cSCC incidence with no tendency for levelling off [2–4]. Following the classical model of multistage carcinogenesis, cSCC is usually manifested as a spectrum of lesions of progressive malignancy. Actinic

**Funding:** This work was supported by the Spanish Government and co-financed by ERDF from the European Union (grants number BFU2014-58429-P and BFU2017-88692-P to JCI), by the Generalitat Valenciana (grant number GVPROMETEO2016-123 to JCI) and Instituto de Salud Carlos III (grant P16/0235 to JS). FH is a recipient of a Predoctoral Fellowship from the Generalitat Valenciana. JS is a Miguel Servet IIA researcher (CPII18/00012). Authors also thank Biobank of the Instituto Valenciano de Oncología (grant number PT17/0015/0051 to JALG from Instituto de Salud Carlos III) for providing the samples analysed in the study. The funders had no role in study design, data collection and analysis, decision to publish, or preparation of the manuscript.

**Competing interests:** The authors have declared that no competing interests exist.

keratosis (AK), which consists in intraepidermal squamous neoplasms, is considered as cSCC precursor but its rate of malignant transformation is unknown. Once developed, the course of carcinoma consists in a local progression through the invasion of deep structures, regional extension by perineural or lymphatic invasion and the development of distant metastasis by vascular invasion. However, very few cases of cSCC reach advanced stages and the factors associated with rapid progression are unknown. In fact, most cSCCs have a good prognosis [5]. Nevertheless, there is a subgroup of SCC coined as high-risk cSCC associated with a higher frequency of local recurrence, lymph node metastasis and significant morbidity and mortality due to patient factors and features of the primary lesion. Risk stratification is critical to the management of cutaneous cSCC because it helps identify patients at high risk of poor outcomes. Although significant progress has been made in defining the clinic and pathologic factors associated with high-risk cSCC, identifying exactly which patients have a higher risk of recurrence, progression, metastasis, and death is not straightforward with these tools [6].

Genetic and genomic studies along the last decade have provided clues on the mutational landscape of cSCC [7–9]. However, the existence of a very high burden of mutation in cSCC, greater than in other solid tumours, makes difficult to identify the precise genetic events and number of mutations required for squamous cell carcinogenesis [10]. Alterations in the *p53* gene are the most common genetic abnormalities found in AK and invasive cSCC, and deregulation of p53 pathway appears to be an early event in carcinogenesis of cSCC [11, 12]. Other molecular alterations in the cell cycle and proliferation regulatory circuit have been identified in cSCC such as loss-of-function mutations in Notch [8, 9, 13] and TFGβ receptors [14], upregulated MAPK [15], amplification of the *c-Myc* locus [16], activating mutations of *ras* [8, 9, 17], aberrant activation of tyrosine kinases EGFR [18] and Fyn [19], inactivation of inositol polyphosphate 5′-phosphatase, a negative regulator of PI3K/Akt signalling [20], as well as reduced expression of PKCδ [21] and $p16^{INK4a}$ $p14^{ARF}$ tumour suppressors [22]. Despite the notable number of genetic studies, the results do not explain the complexity and evolution of the disease. In particular, there is a weak knowledge of the molecular risk factors associated with the high-risk aggressive subtype of cSCC in immunocompetent patients before metastatic events.

Epigenetics, as an emerging field that provides a plausible link between the environment and alterations in gene expression, may be useful to unravel the molecular mechanisms involved in the pathophysiology of this disease [23]. DNA methylation is a particular mechanism of epigenetic regulation that implies a covalent epigenetic modification of cytosines within CpG dinucleotides established and maintained by DNA methyltransferases. The deregulation of the methylome is a hallmark of the development and progression of human cancers [24] and the altered DNA methylation patterns can be used as biomarkers for tumour detection, diagnosis and prognosis [23, 25]. In the case of cSCC, previous studies targeting specific genes have reported promoter hypermethylation associated with the disease in genes such as the cell cycle regulator CDKN2A [22], cadherin CDH1 [26, 27] and CDH13 [28], transcription factor FOXE1 [29], modulators of Wnt signalling SFRPs [30] and FRZB [31], positive regulators of apoptosis ASC [32], G0S2 [33] and DAPK1 [34], and miRNA-204 [35], as well as hypomethylation of the DSS1 gene [36]. However, both DNA hypermethylation and hypomethylation associated with cancer are observed throughout the genome and these alterations, in particular the loss of methylation, may have functional consequences beyond regulation of proximal genes [24]. Due to this, a global vision of epigenetic regulation can only be obtained in large-scale approaches.

In the past decade, the advent of genome-scale analysis techniques, such as Infinium microarray systems, has revolutionized the field and has opened up the new era of epigenomics. These large-scale approaches have been critical to significantly improve our molecular

understanding of multiple diseases and enable potential key insights for developing epigenetic therapies [37, 38]. It has been described the existence of widespread blocks of hypomethylation associated with chronic exposure in older, sun-exposed nonmalignant skin that are maintained in cSCC samples [39]. More recently, it was reported that both AK and cSCC methylation patterns display classical features of cancer methylomes when compared to normal skin; however, no differences between AK and cSCC were detected [40]. Here, we use the last generation Infinium MethylationEPIC BeadChips 850K to interrogate DNA methylation changes in the progression from premalignant actinic keratosis to invasive cSCC, differentiating low-risk and high-risk tumours. The results allowed us to identify methylation signatures able to discriminate between progressive squamous cell carcinoma stages and to provide a prognostic prediction model for disease survival.

## Materials and methods

### Patient samples

The patient samples comprised of 23 skin biopsies from 23 participants diagnosed at the four different stages of cutaneous squamous cell carcinoma (cSCC): actinic keratosis, early invasive carcinoma, high-risk non-metastatic carcinoma and high-risk carcinoma with nodal metastasis (S1 Table). The TNM classification of cSCC was evaluated according to the AJCC (American Joint Committee on Cancer) Cancer Staging Manual (8th edition) [41]. All the patients and samples were recruited from the Departament of Dermatology of the Instituto Valenciano de Oncología (IVO) in Valencia (Spain) after approval by the ethics committee of IVO. Written informed consent for the use of tissue samples was obtained from all included patients or their legal guardians. All research was performed in accordance with relevant guidelines and regulations. The samples were obtained from exposed skin areas from immunocompetent patients. The biopsies were obtained directly from the cSCC tumours at the time of diagnosis by surgical excision of 100–150 mg and stored as fresh frozen tissue, prepared and supplied by the Biobank at IVO. In the case of patients with actinic keratosis, a shave biopsy of the entire lesion was obtained. The samples were divided into two parts, one part being sent to pathology for confirmation of the diagnosis and the other sent for freezing. To carry out the study we proceeded to separate the dermis and epidermis, avoiding the inclusion of dermal parts. This study included only samples of actinic keratosis where a sufficiently large area of epidermal layer could be dissected. In the case of invasive carcinomas, a punch biopsy of the most representative central portion of the tumour was obtained. All samples were evaluated visually by a trained dermatopathologist to validate tumour cellularity. Tumour cells represent around the 80%-90% of total cells in most of the samples with no bias between sample groups.

### DNA extraction

Genomic DNA (gDNA) was extracted from the frozen tissues of the cSCC samples using the QIAcube from Qiagen® and the QIAamp® DNA Investigator Kit (Qiagen®, Germany), according to the manufacturer´s protocol. Prior to performing the methylation studies, a DNA integrity quality control was performed to ensure that the DNA met the standard quality measurements that included a minimum requirement of 200 ng. All DNA samples were treated with RNaseA for 1 h at 45˚C, quantified by the fluorometric method (Quant-iT Pico-Green dsDNA Assay, Life Technologies, CA, USA), and assessed for purity by NanoDrop (Thermo Scientific, MA, USA) 260/280 and 260/230 ratio measurements. DNA integrity of the fresh frozen samples was checked by electrophoresis on a 1.3% agarose gel.

## Epigenomic studies

Epigenomic studies were performed with the Infinium MethylationEPIC BeadChips 850K array (Illumina Inc. USA) used for the interrogation of over 850,000 CpG sites (dinucleotides that are the main target for methylation), which has been previously established as a reliable technology to detect epigenetic alteration [42]. 600 ng of purified DNA were randomly distributed on a 96-well plate and processed using the EZ-96 DNA Methylation kit (Zymo Research Corp., CA, USA) following the manufacturer's recommendations for Infinium assays. The efficiency of bisulfite conversion reaction was checked, being optimal in all the cases with no significant difference between sample groups. Bisulfite-converted DNA (bs-DNA) was processed as previously described [43]. DNA MethylationEPIC beadarray shares the Infinium HD chemistry Assay (Illumina Inc. USA) used to interrogate the cytosine markers with HumanMethylation450 beadchip. Thus, the applicable protocol for MethylationEPIC is the same as for HumanMethylation450, which is the Infinium HD Methylation Assay Protocol. 4 μl of tissue bisulfite-DNA was processed following the Illumina Infinium HD Methylation Assay Protocol, as previously described [43].

The methylation score for each CpG was represented as a beta value according to the fluorescent intensity ratio. Beta values may take any value between 0 (non methylated) and 1 (completely methylated). The raw data (IDAT files) were normalized using functional normalization as implemented in the R-package minfi (version 1.22.1). CpG markers present on Methylation EPIC are classified based on its chromosome location and the feature category gene region as per UCSC annotation (TSS200, TSS1500, 5′UTR, 1st Exon, Body, 3′UTR). This array has a complete coverage of the genome including 99% of genes described. Additional criteria included the location of the marker relative to the CpG substructure (island, shore, shelf, open sea), fantom 5-associated enhancer regions and regulatory regions described on ENCODE project, such as transcription binding site sequences, open chromatin regions and DNase I hypersensitivity clusters.

## Prefiltering

Every beta value in the EPIC array is accompanied by a detection p-value which represents the confidence of a given beta value. Probes and sample filtering involved a two-step process in which unreliable betas with high detection P value P > 0.01 (2220 CpGs) and 2932 CpGs associated with SNPs were removed. Previous analyses indicated that a threshold value of 0.01 allows a clear distinction to be made between reliable and unreliable beta values [44]. Sex chromosome probes were also removed (19681 CpGs). After the filtering, the remaining 841818 CpGs were considered valid for the study.

## Statistical analysis

Two different approaches to assess differential methylation between the different studied groups were followed. First a multinomial elastic net penalized regression was used. Penalized regression methods consist in fitting a regression model subject to a specific restriction (a bound on the value of the coefficients). This method forces the shrinkage of the parameters to zero, potentially performing variable selection at the model-fitting step [45]. Penalization factor for the Elastic Net was selected using 500 repetitions of 10-fold cross-validation. From each repetition the highest lambda at one standard error from the minimum was selected (one-standard-error rule) and the median of the 500 lambda values was used as the final penalization factor. CpGs with coefficients different from zero after the penalization were considered as selected by the analysis. Additionally, a beta regression model was also adjusted for each CpG with each methylation value as response variable and the variable group as predictor [46]. P

values of the beta regression analyses were adjusted for multiple comparisons by using False Discovery Rate. A prognosis model for survival was adjusted using an elastic net penalized cox regression model. Gene Ontology (GO) analysis was performed using *ShinnyGO* [47] with the results of the differential methylation analysis between low-risk initial invasive and high risk non-metastatic and metastatic cutaneous squamous cell carcinoma (cSCC) stages. All statistical analyses were performed using R (version 3.5.1) and R packages *glmnet* (version 2.0–13) and *betareg* (version 3.1–0).

## Results

### Description and classification of multistage cutaneous squamous cell carcinoma patient samples

A critical point in this type of studies is the careful selection and classification of samples at different stages of the evolution of the pathology. In this work the classification is based on the Cancer Staging Manual 8th Edition of the American Joint Committee on Cancer (AJCC) [41], which distinguishes cSCC stages based on the presence or absence of several high risk-features such as the presence of significant perineural invasion, invasion of the tumour beyond subcutaneous fat or depth greater than 6mm and presence of bone involvement. The final cohort included patients with precursors of squamous cell carcinoma (actinic keratosis group) and patients with cSCC at different stages. Among the latter, we distinguish between carcinomas in the early stages of the disease (low-risk initially invasive cSCC group) defined by the absence of high risk features (T1 and T2 stages of AJCC-8) and cases at more advanced stages of the disease defined by the presence of high risk features (T3 and T4 stages of AJCC-8), differentiating those without nodal metastasis at the time of diagnosis (high-risk non metastatic group) from those with nodal metastasis (high-risk metastatic group). Long follow-up of all the cases is available to confirm the actual stage of the disease, and the overall survival. The patient characteristics are summarized in S1 Table.

### Global DNA methylation profile in multistage cutaneous squamous cell carcinoma

In this work, we aimed to determine the overall genome DNA methylation profile in human skin from pre- and post-malignant multistage cSCC using the Infinium DNA Methylation EPIC BeadChip microarray 850K (Illumina), in order to investigate the existence of potential elements of distinction among specific stages of carcinoma development. In a first approach, an exploratory principal component analysis (PCA) revealed a separation among the clusters associated with the four analysed groups (Fig 1A). In particular, the results suggest that major transitions in methylation pattern occur between actinic keratosis, low-risk initial invasive carcinoma stage and high-risk carcinoma stages, with small differences associated with the presence or not of metastasis.

We wondered if DNA methylation alterations in cSCC occur sequentially throughout the different stages of skin tumour. To investigate this point, we performed a penalized ordinal regression including all CpGs as potential predictors. However, no ordinal association was found. This indicates that no evident sequential DNA methylation changes occur in the development of cSCC. Next, we performed a differential methylation study using a beta regression analysis for each CpG with the group as independent variable. Using an FDR-adjusted p-value cut-off of 0.05, we found 272,964 differentially methylated CpGs among the four groups that represent 32% of the CpGs investigated (Fig 1B). A very high number of differentially methylated CpGs between the four groups were still detected when more stringent analysis (p-values

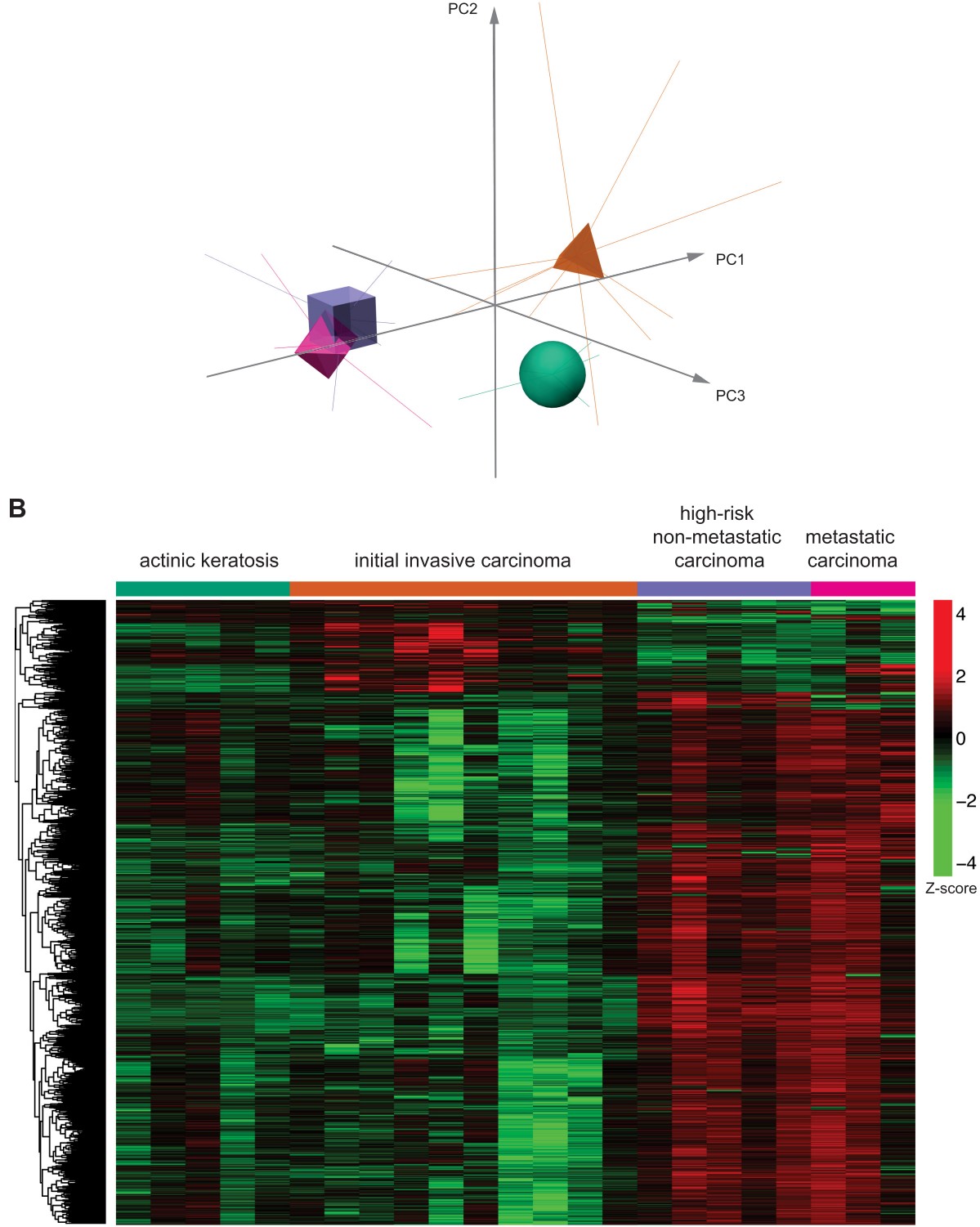

**Fig 1. Analysis of the global DNA-methylation profile in multistage cSCC.** *A)* Representation of the Principal Component Analysis (PCA) on the methylation data. Centroids of each group of patients are represented by different shapes and colors (legend indicated in B). *B)* Heatmap with a random sample of 5000 CpGs from the results of the beta regression analysis for differential methylation among all cutaneous squamous cell carcinoma groups of patients. Z-score colour scale ranges from green for lower methylation to red for higher methylation levels.

from $10^{-2}$ to $10^{-10}$) were carried out (S2 Table). An initial inspection of the heatmap suggests an interesting pattern in which the methylation state decreases from actinic keratosis to low-risk initial invasive carcinoma group but increases from low risk to high risk (non-metastatic and metastatic) groups. These results reveal a massive and complex pattern in the DNA methylation evolution during the development of cSCC affecting the global epigenomic structure.

### Identification of a DNA-methylation signature characterizing the different stages of cSCC

In order to identify the minimum number of CpGs characterizing the distinct groups, we performed an elastic net penalized multinomial logistic regression with stage as response variable and all CpGs as potential predictors of epigenetic changes able to discriminate among groups. Interestingly, the results of this model showed that a profile consisting of only 94 CpG was enough to correctly classify 100% of the samples. The results of this supervised analysis are depicted in a clustered heatmap (Fig 2) and presented in S3 Table. Strikingly, the heatmap of these CpGs suggested that, on one hand, the initial invasive group showed lower methylation levels than premalignant actinic keratosis, and on the other hand, high-risk non-metastatic

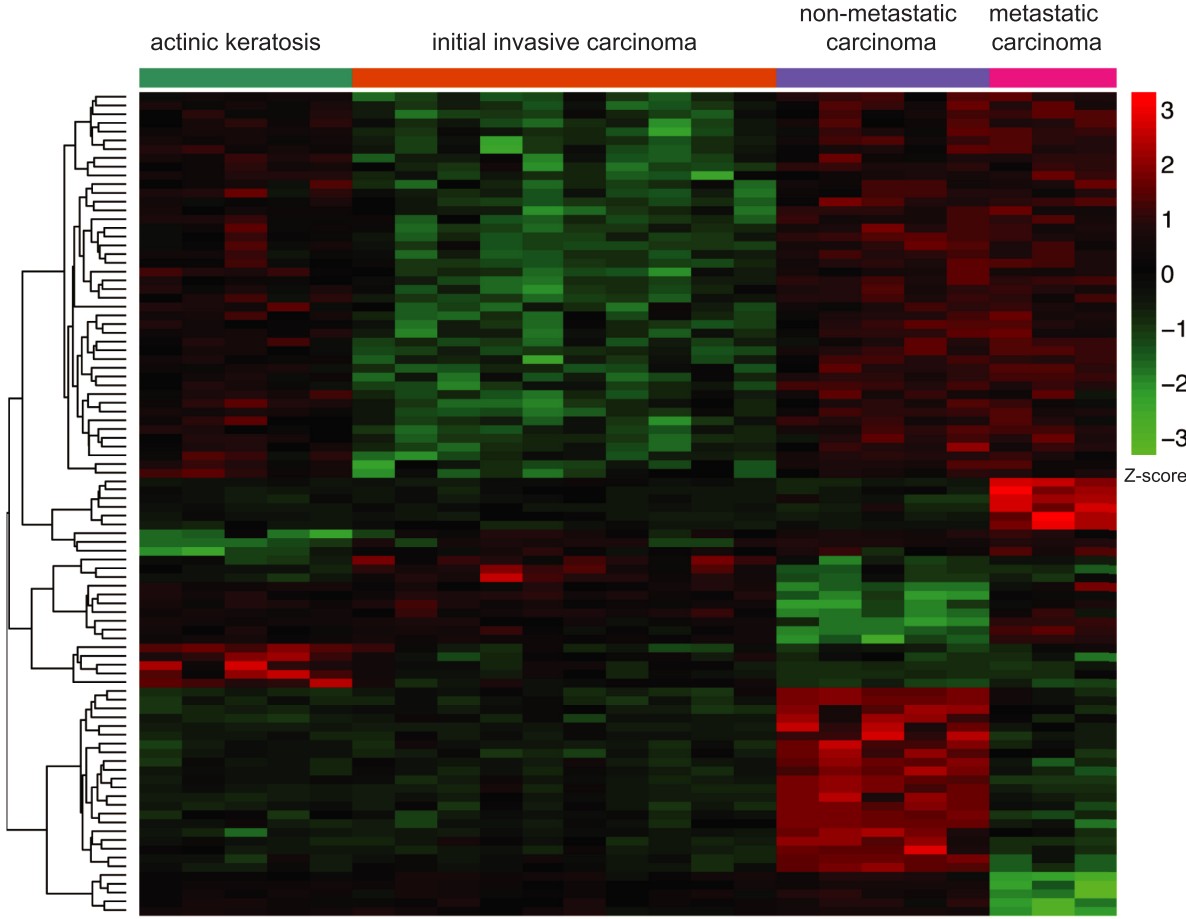

**Fig 2. Identification of a DNA-methylation signature characterizing the different stages of cSCC.** Heatmap with the results of the elastic net analysis for discrimination among all cutaneous squamous cell carcinoma groups of patients. Rows (CpGs) and columns (individuals) are ordered according to the results of a hierarchical clustering algorithm. Z-score colour scale ranges from green for lower methylation to red for higher methylation levels.

and metastatic groups displayed higher methylation levels. This confirms the existence of a non-sequential and complex pattern of DNA-methylation during cSCC evolution. Importantly, we have generated an epigenetic model based on genome wide profiling that may classify prospective samples based on the methylation level of 94 CpGs.

Additionally, we performed a technical validation of the signature, by carrying out a pyrophosphate sequencing with 5 CpGs representative of the 94 CpGs. The results validated the discrimination power of the signature, since the pyrosequencing data was able to correctly classify 82% of the samples (18 out of 22) and achieved a weighted Bangdiwala score of 0.90 [48]. The representation of the confusion matrix as an agreement plot is included as S1 Fig.

## Comparison of DNA-methylation pattern between low-risk and high-risk stages

To better differentiate between cSCC stages depending on the risk, a new elastic net penalized multinomial logistic regression analysis was performed to compare low-risk initial invasive cSCC patients to high-risk non-metastatic and metastatic cSCC patients. Given the small differences we observed in the PCA analysis between the two latter groups of samples (see Fig 1A) and previous reported analysis indicating that there is not widespread differences in the methylation patterns between non-mestastatic versus metastatic cSCC [31], the high-risk non-metastatic and metastatic samples were considered as a single group in this analysis in order to optimize the output. The results revealed 637 differentially methylated CpGs (Fig 3A and S4 Table) between low-risk and high-risk cSCC. Most of them (602 CpGs) represent a gain of methylation in high-risk versus low-risk stages, whereas the minority of cases (35 CpGs) present a loss of methylation in high-risk versus low-risk stages. Thus, and consistent with the analysis showed above, the low-risk invasive group also showed lower median methylation levels than high-risk groups (Fig 3B). In conclusion, there is a risk-dependent change in DNA methylation patterns, mostly indicating a gain of methylation.

Next, the distribution of the identified CpGs with gain and loss of methylation between high-risk versus low-risk stages, was analysed in the Illumina EPIC array regarding functional genomic distribution (FGD) (Fig 3C) and CpG substructure context (Fig 3D). Regarding FGD, CpGs with gain of methylation were preferentially found in the gene body whereas CpGs with loss of methylation were enriched in the promoter regions (Fig 3C). Notably, based on CpG substructure context, the distribution of CpGs with gain of methylation revealed a significant enrichment in open sea (low density CpG regions), while CpGs with loss of methylation were mainly found in CpG shores (Fig 3D). Strikingly, CpGs located in CpG islands (high density CpG regions) were underrepresented in both cases.

Finally, the functional signature of the genes associated with DNA methylation changes was investigated with the ShinnyGO data set [47]. 395 out of the 454 genes with altered CpG methylation were classified based on Biological Process categories. The detected enriched categories using a p-value cutoff of 0.01 are shown in Fig 4 and S5 Table. Interestingly, this analysis revealed that genes with changes in CpG methylation are mainly associated with categories that are directly related to cancer, such as signal transduction and cell motility, as well as phospholipid metabolic processes. Further analysis of KEGG (Kyoto Enciclopedia of Genes and Genomes) pathways [49] showed an enrichment in key signalling pathways related to cell proliferation and cell fate like Notch, hedgehog, MAPK, TFGβ and Wnt pathways (S6 Table).

## Prognosis signature

Finally, we attempted to produce a prognostic prediction model for overall survival in cSCC patients. To do this, a penalized cox regression model using all CpGs as potential predictors

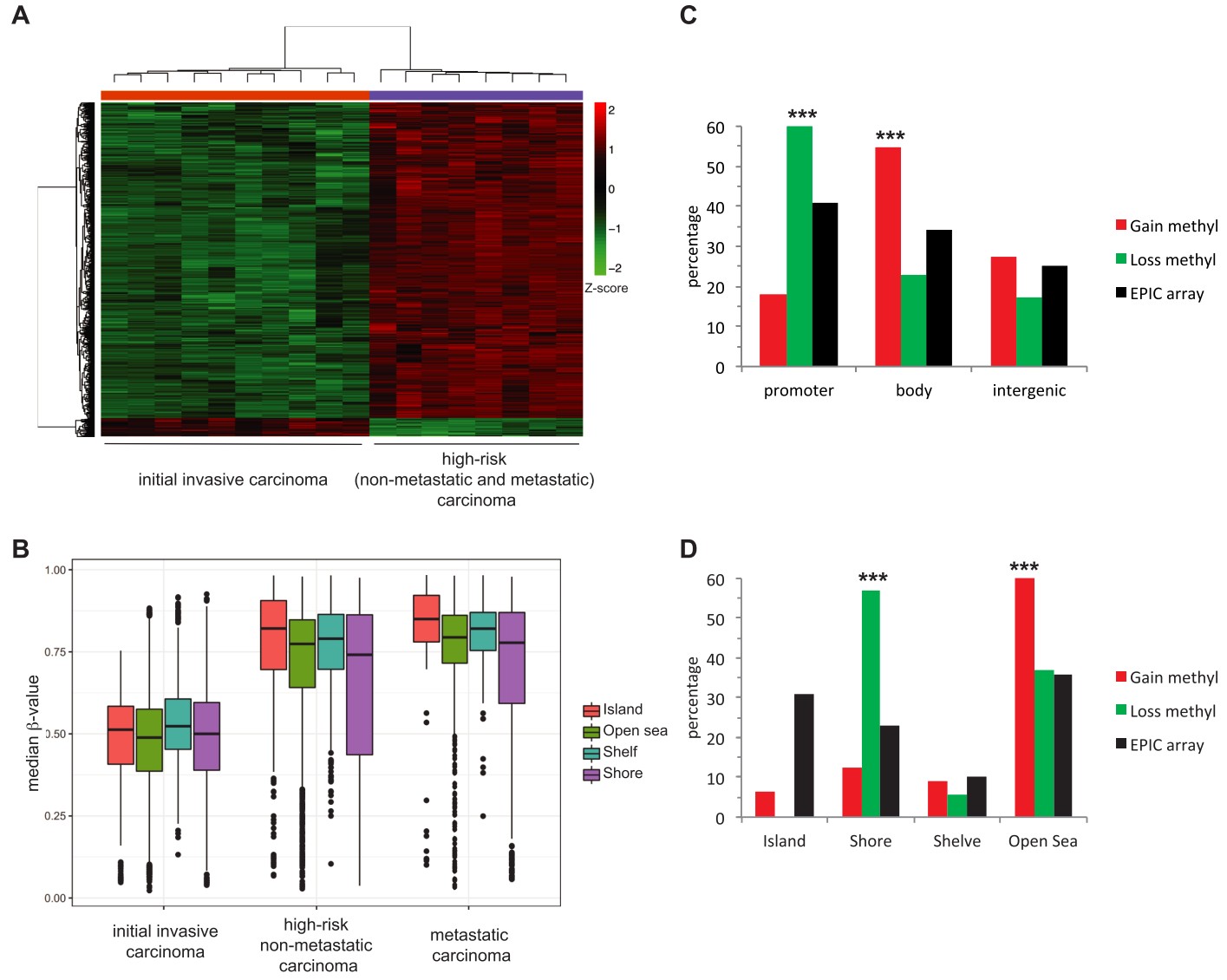

**Fig 3. Comparison of DNA-methylation pattern between low-risk and high-risk stages.** *A)* Heatmap with the results of the elastic net analysis for differential methylation between low-risk initial invasive and high risk non-metastatic and metastatic cutaneous squamous cell carcinoma (cSCC) stages. Z-score colour scale ranges from green for lower methylation to red for higher methylation levels. *B)* DNA methylation median β-values of the different epigenomic substructures in the different groups of sCCC. Box plots indicate highly significant (p < 0.001, two-sided t-test) hypermethylation of all substructures in both high-risk non-metastatic and metastatic groups compared to low-risk initial invasive cSCC samples. *C)* Distribution of selected differentially methylated CpGs in high-risk compared to low-risk cSCC (hypomethylated in green and hypermethylated in red) and total CpG in the Illumina DNA methylation EPIC array in genes, promoters, gene body or intergenic regions. *D)* Distribution of differentially methylated CpGs in CpG islands, shore, shelf or open sea.

was adjusted. The adjusted elastic net model selected a specific signature of 32 CpGs (S7 Table) able to predict risk of death in these patients with a very high discriminant capacity (Area under the curve, AUC = 0.98). This discrimination capacity was represented using model-estimated survival curves showing that patients with a higher predicted risk had a markedly shorter survival time compared to patients with a lower predicted risk (Fig 5). Thus, the analysis of the methylation status of these 32 CpGs emerges as a good approach to predict the evolution of the disease.

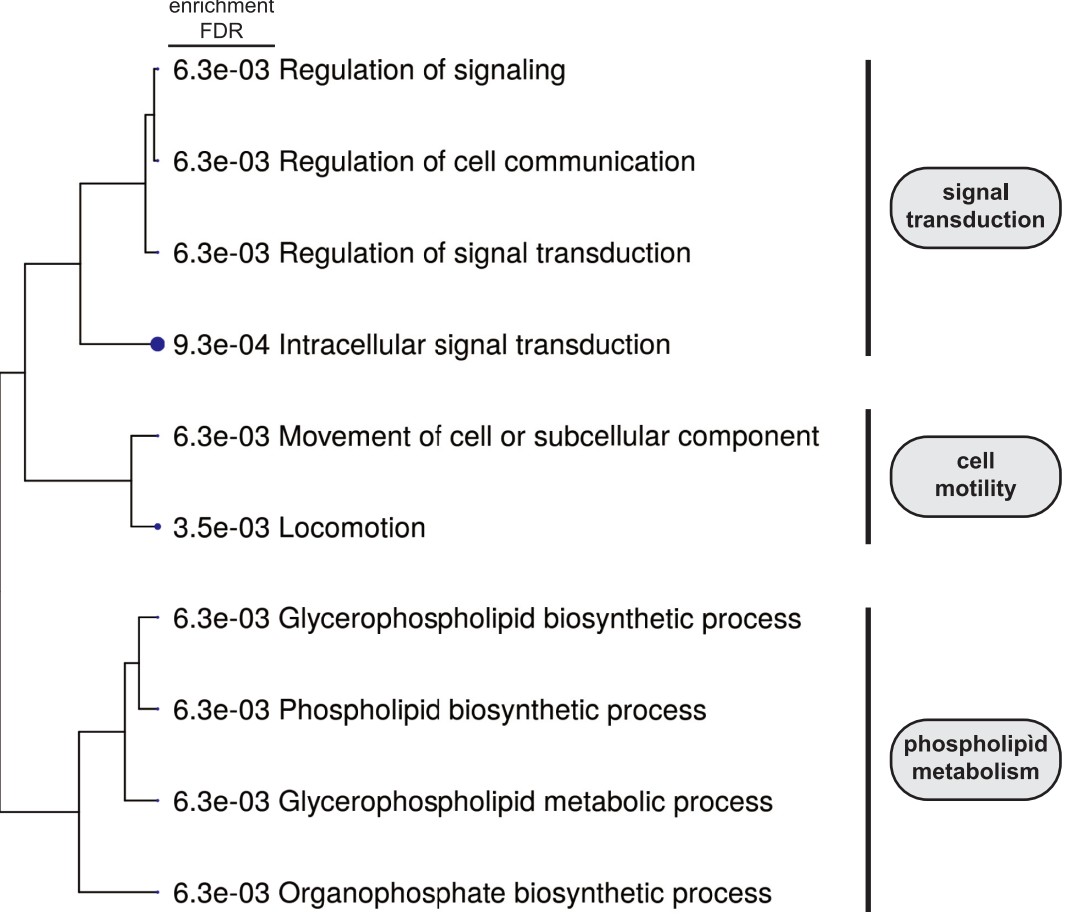

**Fig 4. Analysis of genes associated to differentially methylated CpG between low-risk and high-risk stages.** Hierarchical clustering tree of the enriched gene sets obtained from the GO analysis of the genes selected in the comparison between low-risk initial invasive and high risk non-metastatic and metastatic cutaneous squamous cell carcinoma (cSCC) stages.

## Discussion

In recent years, epigenetics has become a discipline capable of explaining some aspects of different pathologies. In particular, it has shed light on the evolution of different types of cancer [24]. cSCC is a major type of skin cancer with an increasing incidence. However, it has been less studied than other skin cancers and there is not a definitive description of its genetic and epigenetic bases. In this context, this work tried to analyse the epigenetic phenomena underlying the evolution of the pathology.

cSCC is a particular type of cancer in which tumour cells accumulate a big amount of genetic alterations due to chronic exposure to sun radiation[10]. It has been hypothesized that ultraviolet radiation in natural sunlight may promote tumorigenesis not merely due to DNA damage, but also through induction of epigenetic dysregulation. Consistent with this, previous reports have described that aberrant DNA-methylation patterns related to cSCC, when compared to healthy skin, seems to involve vast changes encompassing a large proportion of the genome [39, 40]. Thus, cSCC is associated with massive alterations of both genetic and epigenetic traits. We have now identified that an enormous amount of CpG shows significant changes between the different stages of the disease. This indicates that the massive epigenetic

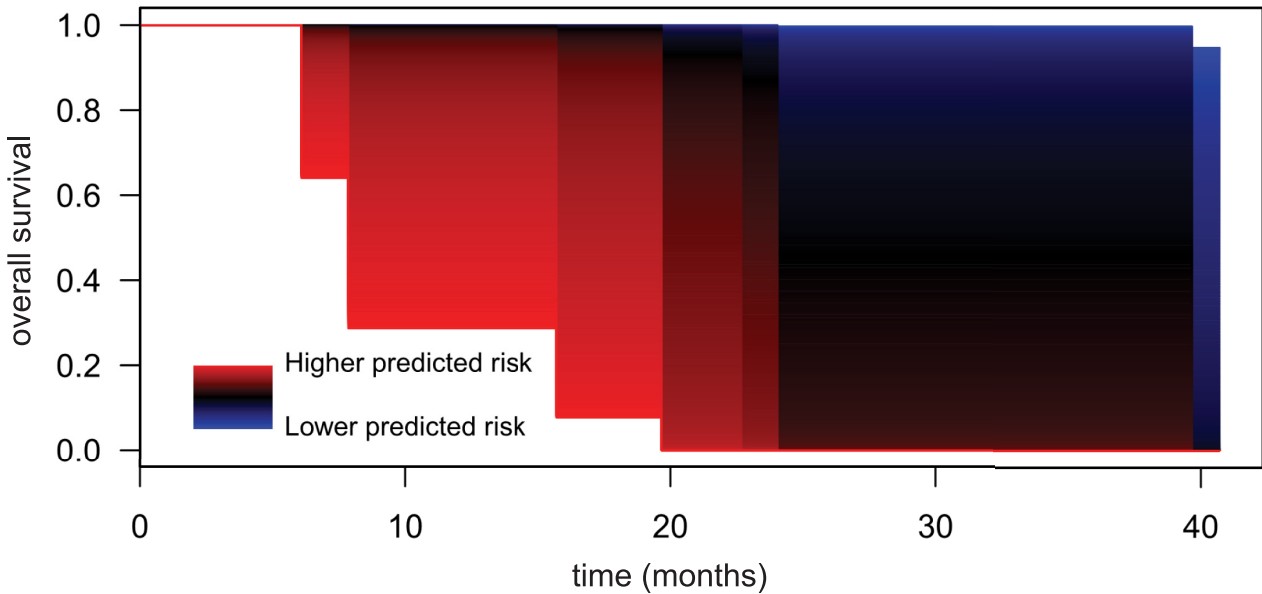

**Fig 5. Survival model based on a prognosis DNA-methylation signature.** Representation of the predicted survival curves in patients with cSCC for different estimated risk values by the regression model using 32 CpGs. Patients with higher predicted risk levels show sizeably lower overall survival (OS) in comparison with patients with lower predicted risk levels.

alterations associated with cSCC is not just a stationary terminal characteristic of tumour cells; rather, enormous changes are dynamically occurring along the evolution of the disease through their different stages. Surprisingly, it seems to exist a non-sequential pattern of DNA-methylation changes during cSCC evolution. What it is clear when the comparison is restricted to low-risk against high-risk samples, is that malignancy is associated to a higher DNA-methylation level. It is tempting to speculate that evolution from premalignant actinic keratosis could follow alternative pathways to evolve to a low-risk stage when DNA-methylation is lowered or a high-risk stage when DNA-methylation is increased. In fact, clinical analysis of patients reveals that not all high-risk cSCC have necessarily passed through a low-risk stage. On the other hand, our results demonstrate that the different stages of cSCC present indeed specific DNA-methylation characteristics that allow to distinguish them. In fact, a minimal epigenetic signature that provides a powerful tool to discriminate between the different stages was obtained using DNA-methylation information from only 94 CpGs. This signature could be useful to classify prospective samples.

From a clinicopathological point of view, the most interesting aspect is the ability to distinguish between a low-risk and a high-risk stage. A direct comparison of these groups provides evidence for the existence of considerable changes in the DNA methylation pattern that are associated with a worse tumour evolution. In fact, 637 differentially methylated CpGs were identified. In 95% of cases, the change implies a hypermethylation of CpGs in the high-risk samples. A recent study suggested that UV-irradiation through chronic sun exposure gives rise to large hypomethylated blocks in healthy epidermis [39]. In view of our results, it is tempting to speculate that DNA-hypomethylation caused by sun exposure characterizes the premalignant and low-risk stages of cSCC and that the evolution to more advanced stages of worse prognosis occurs when the state of methylation is reversed changing from hypo to hypermethylated DNA.

Specific changes in gene function related to cell proliferation, cell fate and cell motility could be epigenetically controlled in the transition from low-risk to high-risk stage. DNA methylation could affect gene function in different ways [24, 50]. The methylation state of a promoter affects gene expression, DNA hypermethylation resulting in gene silencing. Also, DNA methylation increases the mutation rate and tumours tend to be globally hypomethylated, which generally results in genomic instability. In our analysis, CpGs with gain of methylation are preferentially found in the gene body whereas CpGs with loss of methylation were enriched in the promoter regions. It can be speculated that, overall, the loss of methylation in promoters could lead to increased expression of key genes and that hypermethylation in body regions could increase the mutation rate. Regarding the CpG substructure, it is surprising that CpG islands were underrepresented in the group of differentially methylated CpGs between low-risk and high-risk stages. Nevertheless, other authors have also pointed out the relevance of methylation changes in open seas and shores in aged sun-exposed skin and cSCC [39, 40], as occurs in other cancers [51]. Thus, the major changes in the methylome during the development of cSCC does not seem to occur in CpG islands.

Interestingly, the GO analysis of the genes associated with differentially methylated CpG between low-risk and high-risk stages, indicates an enrichment of genes in categories directly involved in cancer development like signal transduction and cell motility. In particular, it has to be noted that the number of genes related to signal transduction that manifested an alteration in their DNA-methylaton pattern is of 119 (S5 Table); this suggest that the transition to a high-risk state of the disease could be caused by a collection of dysfunctions in several signalling processes controlling cell behaviour. In fact, the KEGG analysis identifies important signalling pathways like MAPK, Notch, Wnt, TFGβ and hedgehog pathways, as potential pathways missregulated in the transition to a high-risk stage. It is important to remark that epigenetic alterations in other genes related to the Wnt-signalling pathway had been previously reported in cSCC samples [30, 31], as well as genetic mutation in Notch receptors [8, 9, 13]; our results support a broad extension of epigenetic alterations in the Wnt pathway and point to a combination of genetic and epigenetic alterations in the deregulation of the Notch pathway in cSCC.

Very recently, other authors reported a study of the DNA-methylome in cSCC [39, 40]. They report that both AK and cSCC methylation patterns display classical features of cancer methylomes. Importantly however, and in contrast to our results, no differences in the methylation profile between these stages of the pathology were identified using linear regression methods. A big difference between Rodriguez-Paredes et al. analysis and our analysis is that Rodriguez-Paredes et al. only considered two groups of samples, premalignant AK and tumour samples, independently of the stage of evolution of cSCC. As commented above, we have detected a complex non-linear evolution of the DNA-methylation pattern during the development of cSCC, so it is possible that a low-stringent classification of samples could mask the detection of changes. To address the potential conflict between both studies, we performed an analysis on our dataset including our three cSCC subgroups (low-risk initial invasive, high risk non metastatic and metastatic) as one single group and compared this group against the actinic keratosis group. The results were completely in agreement with those of Rodriguez-Paredes et al, finding no significant differences between the actinic keratosis patients and all-grouped cSCC patients. This result confirmed that disregarding the non-sequential epigenetic pattern among the different cSCC subgroups results in an inability to detect epigenetic alterations, since DNA methylation gain/loss among cSCC subgroups could be compensated and cancelled.

Finally, a prognostic prediction model was developed. Histopathology and immunohistochemistry, the standard methods for cancer diagnosis, have important limitations since they

are time consuming and the result depends on a pathologist's subjective interpretation. Due to this, there is a need for molecular biomarkers that could be useful in tumour diagnosis and grading and with a potential use in prognosis of the disease. In this sense, we have identified a signature based on the methylation state of 32 CpGs that provides a risk-score, which is associated with the overall survival of the patients. Notably, one of these CpGs is located in the gene coding for p73, a functional homolog of p53 that has been related to cancer [52–54]. p53 deregulation is an early event in carcinogenesis of cSCC, so it is tempting to speculate that alteration of its functional homolog p73 could be critical for bad prognosis. Although this 32 CpG signature is a preliminary proposal, it supports the general conclusion that the analysis of the degree of methylation for DNA samples from cSCC patients can be of great value, not only in the diagnosis but also in the prognosis. The development of future tools following this basis and its validation with larger cohorts could be of important help in clinical practice.

## Conclusions

Our results showed massive non-sequential changes in DNA-methylome along the different cSCC stages, identified a minimal methylation signature that discriminates between stages and revealed epigenetic traits characteristic of high-risk tumours. In addition, a prognostic prediction model in cSCC patients identified a methylation signature able to predict the overall survival of patients. Thus, we conclude that DNA-methylation information can be of great value in the diagnosis and the prognosis of cSCC.

## Supporting information

**S1 Fig. Technical validation of the 94CpG signature by pyrophosphate sequencing of 5 representative CpG.** Graph represents the agreement plot derived from providing visual assessment of agreement between observed and predicted values from pyrosequencing data of 5CpG in our cohort as described in Bangdiwala et al. 2008, *Journal of Clinical Epidemiology* 61:866–874. Shaded areas represent cell frequencies from the confusion matrix of the adjusted multinomial model. The degree of agreement is visually expressed by the proportion of area in the darkened squares compared with the total area defined by the row and column marginal totals. A: actinic keratosis; B: low-risk cSCC; C: high-risk non metastatic cSCC; D: high-risk metastatic cSCC. Note that only 22 out of the 23 samples described in the manuscript could be analysed in the pyrosequencing assay.
(PDF)

**S1 Table. List of samples.**
(DOCX)

**S2 Table. Number of differentially methylated CpGs between groups.**
(DOCX)

**S3 Table. DNA-methylation signature: 94 CpGs discriminating the four groups.**
(XLSX)

**S4 Table. Differentially methylated CpGs between high-risk and low-risk carcinomas.**
(XLSX)

**S5 Table. GO categories enriched in genes associated to DM CpGs between low-risk and high-risk carcinomas.**
(XLSX)

**S6 Table. Enriched KEGG pathways.**
(DOCX)

**S7 Table. DNA-methylation prognosis signature: 32 CpGs predicting bad prognosis.**
(XLSX)

## Acknowledgments

We thank Diana García from IIS La Fe and Marta Ramírez-Calvo and Arantxa Rodríguez-Hernández from IVO for their technical support.

## Author Contributions

**Conceptualization:** Onofre Sanmartín, Jose Antonio López-Guerrero, M. Carmen Bañó, J. Carlos Igual, Inma Quilis, Juan Sandoval.

**Data curation:** David Hervás-Marín, J. Carlos Igual, Inma Quilis, Juan Sandoval.

**Formal analysis:** David Hervás-Marín, J. Carlos Igual, Inma Quilis.

**Funding acquisition:** J. Carlos Igual.

**Investigation:** David Hervás-Marín, Faatiemah Higgins, Inma Quilis, Juan Sandoval.

**Methodology:** David Hervás-Marín, Faatiemah Higgins, Juan Sandoval.

**Project administration:** J. Carlos Igual, Inma Quilis, Juan Sandoval.

**Resources:** Onofre Sanmartín, Jose Antonio López-Guerrero, Juan Sandoval.

**Software:** David Hervás-Marín.

**Supervision:** Onofre Sanmartín, Jose Antonio López-Guerrero, M. Carmen Bañó, J. Carlos Igual, Inma Quilis, Juan Sandoval.

**Validation:** Onofre Sanmartín, J. Carlos Igual, Inma Quilis, Juan Sandoval.

**Writing – original draft:** David Hervás-Marín, Onofre Sanmartín, J. Carlos Igual, Inma Quilis, Juan Sandoval.

**Writing – review & editing:** Onofre Sanmartín, M. Carmen Bañó, J. Carlos Igual, Inma Quilis, Juan Sandoval.

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
