## [Decision Letter · Decision Letter 0]

14 Oct 2019

PONE-D-19-25692

Genome wide DNA methylation profiling identifies specific epigenetic features in high-risk cutaneous squamous cell carcinoma.

PLOS ONE

Dear Dr. Igual,

Thank you for submitting your manuscript to PLOS ONE. After careful consideration, we feel that it has merit but does not fully meet PLOS ONE’s publication criteria as it currently stands. Therefore, we invite you to submit a revised version of the manuscript that addresses all the points raised during the review process.

As stated in reviewers’ comments, both reviewers suggested multiple improvements for the revision of the manuscript. Specifically, the Editor felt that all the CpG methylation data was obtained by only one technical platform. Thus, Editor thought that the authors should confirm their CpG methylation results of at least some representative CpG sites (especially the signature list of 32 CpGs), using other methylation analysis techniques such as bisulfite sequencing/pyro-sequencing/or methylation-specific PCR.

We would appreciate receiving your revised manuscript by Nov 28 2019 11:59PM. To enhance the reproducibility of your results, we recommend that if applicable you deposit your laboratory protocols in protocols.io, where a protocol can be assigned its own identifier (DOI) such that it can be cited independently in the future. For instructions see: http://journals.plos.org/plosone/s/submission-guidelines#loc-laboratory-protocols

We look forward to receiving your revised manuscript.

Kind regards,

Qian Tao

Academic Editor

PLOS ONE

Journal Requirements:

3. We note that you are reporting an analysis of a microarray, next-generation sequencing, or deep sequencing data set. PLOS requires that authors comply with field-specific standards for preparation, recording, and deposition of data in repositories appropriate to their field. Please upload these data to a stable, public repository (such as ArrayExpress, Gene Expression Omnibus (GEO), DNA Data Bank of Japan (DDBJ), NCBI GenBank, NCBI Sequence Read Archive, or EMBL Nucleotide Sequence Database (ENA)). In your revised cover letter, please provide the relevant accession numbers that may be used to access these data. For a full list of recommended repositories, see http://journals.plos.org/plosone/s/data-availability#loc-omics or http://journals.plos.org/plosone/s/data-availability#loc-sequencing.

Reviewers' comments:

Reviewer's Responses to Questions

**Comments to the Author**

1. Is the manuscript technically sound, and do the data support the conclusions?

Reviewer #1: Partly

Reviewer #2: Yes

2. Has the statistical analysis been performed appropriately and rigorously? 

Reviewer #1: No

Reviewer #2: Yes

3. Have the authors made all data underlying the findings in their manuscript fully available?

Reviewer #1: Yes

Reviewer #2: Yes

4. Is the manuscript presented in an intelligible fashion and written in standard English?

Reviewer #1: Yes

Reviewer #2: Yes

5. Review Comments to the Author

Reviewer #1: A methylation profiling study was performed in 23 cases using the Illumina EPIC beadchip to identify the methylation changes specific to high-risk cSCC. The samples are carefully selected and classified into 4 groups as premalignant actinic keratosis, low-risk invasive, high-risk non-metastatic and metastatic cSCC. A large number of differentially methylated CpG sites were found among 4 groups by the regression analysis. The methylation signature including 94 CpG sites to classify different stages was identified and the authors attempted to build a prediction model for overall survival in cSCC. The strength of the study is the use of carefully classified samples with detailed analysis, while the weakness is also obvious with limited number of samples. The reviewer has several major comments that need to be further clarified and addressed as below.

1. What is the bisulfite conversion efficiency for the EPIC beachip? Are they similar among 4 groups?

2. A large number of differentially methylated CpG sites was identified among four groups. The cellularity for these samples is not presented in the manuscript, so it is difficult to evaluate if different carcinoma contents potentially confound this analysis.

3. The author reported that there are no sequential DNA methylation changes occur in the development of sSCC. Only 23 samples were included in the analysis. Is this sample size adequate to draw this conclusion?

4. Considering only 94 selected CpG sites, the authors found that the methylation level is lower in the initial invasive group than premalignant actinic keratosis, while it is higher in the high-risk groups compared to the low-risk group. Does this trend also true for the global methylation level?

5. A prognostic prediction model was established considering all CpGs as potential predictors. Again, the sample size is a limitation. The rational for establishing the prediction model is unclear. It is highly possible that the prognostic value of this signature comes from the association with stage. If it is the case, then, measurement of methylation biomarkers with additional cost does not add any clinical value in practice.

Reviewer #2: This study investigated the DNA methylation changes in patients with different disease stages, from premalignant AK to low-risk invasive cSCC, high-risk non-metastatic and metastatic cSCC. The identified methylation signatures could discriminate different disease stages, but also predict overall survival, providing insights for the diagnosis and prognosis of this disease. However, there are several minor comments of this paper.

1. The methylation analysis showed that the low-risk invasive cSCC exhibited lower methylation levels than premalignant AK, and high-risk non-metastatic and metastatic cSCC exhibited higher methylation levels, indicating a non-sequential and complex pattern of DNA-methylation during cSCC evolution. This is an interesting finding, the authors would be advised to discuss the possible cause of this changes.

2. All of the data are based on microarray analysis, it is better to select several genes for MSP or pyrophosphate sequencing validation, which would support their findings and conclusions.

6. PLOS authors have the option to publish the peer review history of their article (what does this mean?). If published, this will include your full peer review and any attached files.

Reviewer #1: No

Reviewer #2: No

---

## [Author Response · Author response to Decision Letter 0]

21 Nov 2019

RESPONSE TO REVIEWERS

PONE-D-19-25692

Genome wide DNA methylation profiling identifies specific epigenetic features in high-risk cutaneous squamous cell carcinoma.

Response to Reviewer 1

We agree with the reviewer’s comment regarding the limited number of samples included in the study. Obviously, in this kind of studies the more samples the better. However, we would like to emphasize that other cSCC studies with a similar strategy, that is, the use of epigenomics arrays, used a comparable number of samples. Vandiver et al Genome Biol. 2015;16:80 analyzed 7 cSCC samples, whereas Rodriguez-Paredes M et al. Nat Commun. 2018;9(1):577 characterized 16 actinic keratosis samples against 18 cSCC samples. Moreover, none of these studies classify samples in four different stages, which, as the Reviewer kindly comment, is the strength of our study.

Major comments have been addressed as follow:

1. What is the bisulfite conversion efficiency for the EPIC beachip? Are they similar among 4 groups? 

As a part of our standard protocol, we checked the bisulfite conversion reactions analyzing the BS Conversion I and BS Conversion II control probes as described in Illumina user guide. In all the samples, the efficiency of bisulfite conversion was optimal with no significant differences among sample groups. We have introduced a sentence in the Methods section (page 6) to clarify this point.

2. A large number of differentially methylated CpG sites was identified among four groups. The cellularity for these samples is not presented in the manuscript, so it is difficult to evaluate if different carcinoma contents potentially confound this analysis.

It was mentioned in the Materials and Methods section, at the end of Patient samples block, that ‘All samples were evaluated visually by a trained dermatopathologist to validate tumour cellularity’. Indeed, tumour cells in most of the samples represent around a 80%-90% of total cells, and no bias between groups was observed. This data is now included in the text (page 5) to avoid confusion. 

3. The author reported that there are no sequential DNA methylation changes occur in the development of sSCC. Only 23 samples were included in the analysis. Is this sample size adequate to draw this conclusion?

We agree with the reviewer that the statistical power is not enough to rule out the existence of specific and individual CpGs sequential changes. However, our ordinal regression analysis, which assumed changes to be sequential from actinic keratosis to high-risk metastatic cSCC, did not find any CpG able to discriminate among groups. In contrast, the multinomial regression analysis, which was not restricted by the sequential effects assumption, found clear methylation patterns discriminating among the four groups. Considering the reviewer’s suggestion, we have softened the statement regarding the sequential methylation changes. Now it is stated in page 9 that: ‘This indicates that no evident sequential DNA methylation changes occur in the development of cSCC.’

4. Considering only 94 selected CpG sites, the authors found that the methylation level is lower in the initial invasive group than premalignant actinic keratosis, while it is higher in the high-risk groups compared to the low-risk group. Does this trend also true for the global methylation level?

As stated in the previous response, the statistical power is not large enough to draw strong global conclusions regarding methylation patterns, in contrast to the 94 CpG signature which clearly discriminates between the four groups and clearly shows the mentioned pattern. Nevertheless, when examining random subsamples of the whole dataset in a heatmap (Figure 1B), the pattern showed always this same behavior. This is in fact stated in the text in the first paragraph of page 10.

5. A prognostic prediction model was established considering all CpGs as potential predictors. Again, the sample size is a limitation. The rational for establishing the prediction model is unclear. It is highly possible that the prognostic value of this signature comes from the association with stage. If it is the case, then, measurement of methylation biomarkers with additional cost does not add any clinical value in practice.

In this approach we wanted to emphasize the potential role of epigenomics not only in diagnosis but also in prognosis. We agree with the reviewer that the sample size is a limitation and further validation of this model should be performed in future studies. Nevertheless, the model was fitted with an advanced statistical method, elastic net, which is specifically suited for dealing with data sets with a large number of variables and very few observations. 

Regarding the possibility of an association with stage, we assessed the capability of each of the included CpGs in the model in discriminating between the four stages. Remarkably, no CpG showed discriminative power among stages, with the lowest p-value being 0.24. Also, a cox regression model including stage as predictor showed an AUC of ‘only’ 0.72, compared to the AUC of 0.98 of our model including 32 CpGs. Therefore, no evidence suggests that this epigenetic signature might be confounded by stage, which reinforces its value as a potential biomarker for survival prognosis.

 

RESPONSE TO REVIEWERS

PONE-D-19-25692

Genome wide DNA methylation profiling identifies specific epigenetic features in high-risk cutaneous squamous cell carcinoma.

Response to Reviewer 2

1. The methylation analysis showed that the low-risk invasive cSCC exhibited lower methylation levels than premalignant AK, and high-risk non-metastatic and metastatic cSCC exhibited higher methylation levels, indicating a non-sequential and complex pattern of DNA-methylation during cSCC evolution. This is an interesting finding, the authors would be advised to discuss the possible cause of this changes.

It is not obvious the significance of this non-sequential pattern of DNA-methylation during cSCC evolution. What it is clear when the comparison is restricted to low-risk against high-risk samples, is that malignancy is associated to a higher DNA-methylation level. It is tempting to speculate that evolution from premalignant actinic keratosis could follow alternative pathways to evolve to a low-risk stage when DNA-methylation is lowered or a high-risk stage when DNA-methylation is increased. In fact, clinical analysis of patients reveals that not all high-risk cSCC have necessarily passed through a low-risk stage. This hypothesis is now commented in the Discussion section in page 14.

2. All of the data are based on microarray analysis, it is better to select several genes for MSP or pyrophosphate sequencing validation, which would support their findings and conclusions.

We have carried out a pyrophosphate sequencing with 5 CpG of the 94 CpG signature discriminating between the four different stages. The results validated the discrimination power of the signature, since the pyrosequencing data was able to correctly classify 82% of the samples (18 out of 22) by using only 5 CpGs out of the 94 CpGs from the original signature and achieved a weighted Bangdiwala score of 0.90 out of 1 in the agreement test (Bangdiwala et al 2008,The agreement chart as an alternative to the receiver-operating characteristic curve for diagnostic tests. Journal of Clinical Epidemiology, 61 (9), 866-874). These results are now commented in the Results section in page 11 and the representation of the confusion matrix as an agreement plot is included as Supplementary Figure 1.

---

## [Decision Letter · Decision Letter 1]

2 Dec 2019

PONE-D-19-25692R1

Genome wide DNA  methylation profiling identifies specific epigenetic features in high-risk cutaneous squamous cell carcinoma.

PLOS ONE

Dear Dr. Igual,

Thank you for submitting your manuscript to PLOS ONE. After careful consideration, we feel that it has merit but does not fully meet PLOS ONE’s publication criteria as it currently stands. Therefore, we invite you to submit a revised version of the manuscript that addresses all the points raised by the reviewers.

We would appreciate receiving your revised manuscript by Jan 16 2020 11:59PM. To enhance the reproducibility of your results, we recommend that if applicable you deposit your laboratory protocols in protocols.io, where a protocol can be assigned its own identifier (DOI) such that it can be cited independently in the future. For instructions see: http://journals.plos.org/plosone/s/submission-guidelines#loc-laboratory-protocols

We look forward to receiving your revised manuscript.

Kind regards,

Qian Tao

Academic Editor

PLOS ONE

Reviewers' comments:

Reviewer's Responses to Questions

**Comments to the Author**

1. If the authors have adequately addressed your comments raised in a previous round of review and you feel that this manuscript is now acceptable for publication, you may indicate that here to bypass the “Comments to the Author” section, enter your conflict of interest statement in the “Confidential to Editor” section, and submit your "Accept" recommendation.

Reviewer #1: All comments have been addressed

Reviewer #2: All comments have been addressed

2. Is the manuscript technically sound, and do the data support the conclusions?

Reviewer #1: Yes

Reviewer #2: Yes

3. Has the statistical analysis been performed appropriately and rigorously? 

Reviewer #1: Yes

Reviewer #2: Yes

4. Have the authors made all data underlying the findings in their manuscript fully available?

Reviewer #1: Yes

Reviewer #2: Yes

5. Is the manuscript presented in an intelligible fashion and written in standard English?

Reviewer #1: No

Reviewer #2: Yes

6. Review Comments to the Author

Reviewer #1: The author added a statement that "the efficiency of bisulfite conversion reaction was checked, being optimal in all the cases with no significance difference between sample groups". Please clearly specify what bisulfite conversion efficiency was achieved in this study.

There are some grammatical errors in the manuscript. The authors should read through the manuscript carefully and make the correction. For example, it should be ‘no significant difference’ instead of ‘no significance difference’ at line 121 on page 6; it should be ‘distinguishes’ instead of ‘distinguish’ at line 174 on page 8.

.

Reviewer #2: The authors have addressed all my comments, I have no further comments. I recommend this paper to be pulished.

7. PLOS authors have the option to publish the peer review history of their article (what does this mean?). If published, this will include your full peer review and any attached files.

Reviewer #1: No

Reviewer #2: No

---

## [Author Response · Author response to Decision Letter 1]

4 Dec 2019

RESPONSE TO REVIEWER

PONE-D-19-25692

Genome wide DNA methylation profiling identifies specific epigenetic features in high-risk cutaneous squamous cell carcinoma.

Response to Reviewer 1

 We are extending the response regarding Reviewer’s concern about bisulfite conversion since it seems it was not clearly answered. As we stated in our previous response, we checked our Illumina´s internal controls as a routine step in standard protocols in methylation analysis. The analysis of BS Conversion I and BS Conversion II control probes were in the standard parameters recommended by Illumina user guide. We provide now the graphs with this information from Illumina examples and from our experiment to be checked by the Reviewer. We hope this information helps to clarify this point. We would like to add that unless new or modified Illumina protocols are described in the studies, these Illumina´s internal controls are not usually included in manuscripts. Therefore, we have not included them in our manuscript, although we can do it if the Reviewer considers it is necessary.

 This is the text that GenomeStudio® Methylation Module v1.8 User Guide indicates for bisulfite conversion controls. We would like to clarify that Illumina did not update this guide when the Infinium EPIC DNA methylation beadchip array used in this study was released. Therefore, controls in our array are quite similar but not identical to the showed in the user guide. 

 (SEE ATTACHED FILE FOR FIGURES)

And these are the results for our samples:

 Bisulfite Conversion I controls

 Bisulfite Conversion II controls

 (SEE ATTACHED FILE FOR FIGURES)

Thanks for indicating the presence of grammatical errors. A native English speaker author had edited writing and now another independent native English speaker has edited it.

---

## [Editor Report · Decision Letter 2]

10 Dec 2019

Genome wide DNA methylation profiling identifies specific epigenetic features in high-risk cutaneous squamous cell carcinoma

PONE-D-19-25692R2

Dear Dr. Igual,

We are pleased to inform you that your manuscript has been judged scientifically suitable for publication and will be formally accepted for publication once it complies with all outstanding technical requirements.

With kind regards,

Qian Tao

Academic Editor

PLOS ONE
---

## [Editor Report · Acceptance letter]

13 Dec 2019

PONE-D-19-25692R2 

Genome wide DNA methylation profiling identifies specific epigenetic features in high-risk cutaneous squamous cell carcinoma 

Dear Dr. Igual:

I am pleased to inform you that your manuscript has been deemed suitable for publication in PLOS ONE. Congratulations! Your manuscript is now with our production department. 

With kind regards,

on behalf of

Dr. Qian Tao 

Academic Editor

PLOS ONE